# Diffusion Convolutional Recurrent Neural Network: Data-Driven Traffic Forecasting

**Yaguang Li[†], Rose Yu[‡], Cyrus Shahabi[†], Yan Liu[†]**
[†] University of Southern California, [‡] California Institute of Technology
[†] {yaguang, shahabi, yanliu.cs}@usc.edu, [‡] rose@caltech.edu

## Abstract

Spatiotemporal forecasting has various applications in neuroscience, climate and transportation domain. Traffic forecasting is one canonical example of such learning task. The task is challenging due to (1) complex spatial dependency on road networks, (2) non-linear temporal dynamics with changing road conditions and (3) inherent difficulty of long-term forecasting. To address these challenges, we propose to model the traffic flow as a diffusion process on a directed graph and introduce *Diffusion Convolutional Recurrent Neural Network* (DCRNN), a deep learning framework for traffic forecasting that incorporates both spatial and temporal dependency in the traffic flow. Specifically, DCRNN captures the spatial dependency using bidirectional random walks on the graph, and the temporal dependency using the encoder-decoder architecture with scheduled sampling. We evaluate the framework on two real-world large scale road network traffic datasets and observe consistent improvement of 12% - 15% over state-of-the-art baselines.

## 1 Introduction

Spatiotemporal forecasting is a crucial task for a learning system that operates in a dynamic environment. It has a wide range of applications from autonomous vehicles operations, to energy and smart grid optimization, to logistics and supply chain management. In this paper, we study one important task: traffic forecasting on road networks, the core component of the intelligent transportation systems. The goal of traffic forecasting is to predict the future traffic speeds of a sensor network given historic traffic speeds and the underlying road networks.

This task is challenging mainly due to the complex spatiotemporal dependencies and inherent difficulty in the long term forecasting. On the one hand, traffic time series demonstrate strong *temporal dynamics*. Recurring incidents such as rush hours or accidents can cause non-stationarity, making it difficult to forecast long-term. On the other hand, sensors on the road network contain complex yet unique *spatial correlations*. Figure 1 illustrates an example. Road 1 and road 2 are correlated, while road 1 and road 3 are not. Although road 1 and road 3 are close in the Euclidean space, they demonstrate very different behaviors. Moreover, the future traffic speed is influenced more by the downstream traffic than the upstream one. This means that the spatial structure in traffic is non-Euclidean and directional.

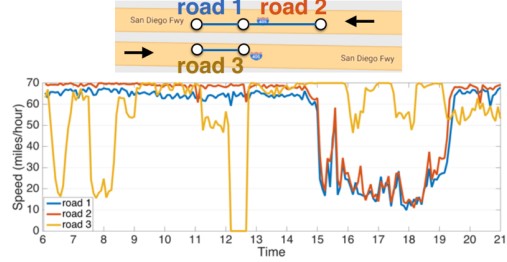

Figure 1: Spatial correlation is dominated by road network structure. (1) Traffic speed in road 1 are similar to road 2 as they locate in the same highway. (2) Road 1 and road 3 locate in the opposite directions of the highway. Though close to each other in the Euclidean space, their road network distance is large, and their traffic speeds differ significantly.

Traffic forecasting has been studied for decades, falling into two main categories: knowledge-driven approach and data-driven approach. In transportation and operational research, knowledge-driven methods usually apply queuing theory and simulate user behaviors in traffic (Cascetta, 2013). In time series community, data-driven methods such as Auto-Regressive Integrated Moving Average (ARIMA) model and Kalman filtering remain popular (Liu et al., 2011; Lippi et al., 2013). However, simple time series models usually rely on the stationarity assumption, which is often violated by

the traffic data. Most recently, deep learning models for traffic forecasting have been developed in Lv et al. (2015); Yu et al. (2017b), but without considering the spatial structure. Wu & Tan (2016) and Ma et al. (2017) model the spatial correlation with Convolutional Neural Networks (CNN), but the spatial structure is in the Euclidean space (e.g., 2D images). Bruna et al. (2014), Defferrard et al. (2016) studied graph convolution, but only for undirected graphs.

In this work, we represent the pair-wise spatial correlations between traffic sensors using a directed graph whose nodes are sensors and edge weights denote proximity between the sensor pairs measured by the road network distance. We model the dynamics of the traffic flow as a diffusion process and propose the *diffusion convolution* operation to capture the spatial dependency. We further propose *Diffusion Convolutional Recurrent Neural Network* (DCRNN) that integrates *diffusion convolution*, the *sequence to sequence* architecture and the *scheduled sampling* technique. When evaluated on real-world traffic datasets, DCRNN consistently outperforms state-of-the-art traffic forecasting baselines by a large margin. In summary:

- We study the traffic forecasting problem and model the spatial dependency of traffic as a diffusion process on a directed graph. We propose *diffusion convolution*, which has an intuitive interpretation and can be computed efficiently.

- We propose *Diffusion Convolutional Recurrent Neural Network* (DCRNN), a holistic approach that captures both spatial and temporal dependencies among time series using *diffusion convolution* and the sequence to sequence learning framework together with scheduled sampling. DCRNN is not limited to transportation and is readily applicable to other spatiotemporal forecasting tasks.

- We conducted extensive experiments on two large-scale real-world datasets, and the proposed approach obtains significant improvement over state-of-the-art baseline methods.

## 2 METHODOLOGY

We formalize the learning problem of spatiotemporal traffic forecasting and describe how to model the dependency structures using *diffusion convolutional recurrent neural network*.

### 2.1 TRAFFIC FORECASTING PROBLEM

The goal of traffic forecasting is to predict the future traffic speed given previously observed traffic flow from $N$ correlated sensors on the road network. We can represent the sensor network as a weighted directed graph $\mathcal{G} = (\mathcal{V}, \mathcal{E}, \boldsymbol{W})$, where $\mathcal{V}$ is a set of nodes $|\mathcal{V}| = N$, $\mathcal{E}$ is a set of edges and $\boldsymbol{W} \in \mathbb{R}^{N \times N}$ is a weighted adjacency matrix representing the nodes proximity (e.g., a function of their road network distance). Denote the traffic flow observed on $\mathcal{G}$ as a graph signal $\boldsymbol{X} \in \mathbb{R}^{N \times P}$, where $P$ is the number of features of each node (e.g., velocity, volume). Let $\boldsymbol{X}^{(t)}$ represent the graph signal observed at time $t$, the traffic forecasting problem aims to learn a function $h(\cdot)$ that maps $T'$ historical graph signals to future $T$ graph signals, given a graph $\mathcal{G}$:

$$[\boldsymbol{X}^{(t-T'+1)}, \cdots, \boldsymbol{X}^{(t)}; \mathcal{G}] \xrightarrow{h(\cdot)} [\boldsymbol{X}^{(t+1)}, \cdots, \boldsymbol{X}^{(t+T)}]$$

### 2.2 SPATIAL DEPENDENCY MODELING

We model the spatial dependency by relating traffic flow to a diffusion process, which explicitly captures the stochastic nature of traffic dynamics. This diffusion process is characterized by a random walk on $\mathcal{G}$ with restart probability $\alpha \in [0, 1]$, and a state transition matrix $\boldsymbol{D}_O^{-1}\boldsymbol{W}$. Here $\boldsymbol{D_O} = \text{diag}(\boldsymbol{W}\boldsymbol{1})$ is the out-degree diagonal matrix, and $\boldsymbol{1} \in \mathbb{R}^N$ denotes the all one vector. After many time steps, such Markov process converges to a stationary distribution $\boldsymbol{\mathcal{P}} \in \mathbb{R}^{N \times N}$ whose $i$th row $\boldsymbol{\mathcal{P}}_{i,:} \in \mathbb{R}^N$ represents the likelihood of diffusion from node $v_i \in \mathcal{V}$, hence the proximity w.r.t. the node $v_i$. The following Lemma provides a closed form solution for the stationary distribution.

**Lemma 2.1.** *(Teng et al., 2016) The stationary distribution of the diffusion process can be represented as a weighted combination of infinite random walks on the graph, and be calculated in closed form:*

$$\boldsymbol{\mathcal{P}} = \sum_{k=0}^{\infty} \alpha(1-\alpha)^k \left(\boldsymbol{D}_O^{-1}\boldsymbol{W}\right)^k \tag{1}$$

where $k$ is the diffusion step. In practice, we use a finite $K$-step truncation of the diffusion process and assign a trainable weight to each step. We also include the reversed direction diffusion process,

such that the bidirectional diffusion offers the model more flexibility to capture the influence from both the upstream and the downstream traffic.

**Diffusion Convolution**    The resulted diffusion convolution operation over a graph signal $\boldsymbol{X} \in \mathbb{R}^{N \times P}$ and a filter $f_{\boldsymbol{\theta}}$ is defined as:

$$\boldsymbol{X}_{:,p} \star_{\mathcal{G}} f_{\boldsymbol{\theta}} = \sum_{k=0}^{K-1} \left( \theta_{k,1} \left( \boldsymbol{D}_O^{-1} \boldsymbol{W} \right)^k + \theta_{k,2} \left( \boldsymbol{D}_I^{-1} \boldsymbol{W}^{\intercal} \right)^k \right) \boldsymbol{X}_{:,p} \quad \text{for } p \in \{1, \cdots, P\} \quad (2)$$

where $\boldsymbol{\theta} \in \mathbb{R}^{K \times 2}$ are the parameters for the filter and $\boldsymbol{D}_O^{-1} \boldsymbol{W}, \boldsymbol{D}_I^{-1} \boldsymbol{W}^{\intercal}$ represent the transition matrices of the diffusion process and the reverse one, respectively. In general, computing the convolution can be expensive. However, if $\mathcal{G}$ is sparse, Equation 2 can be calculated efficiently using $O(K)$ recursive sparse-dense matrix multiplication with total time complexity $O(K|\mathcal{E}|) \ll O(N^2)$. See Appendix B for more detail.

**Diffusion Convolutional Layer**    With the convolution operation defined in Equation 2, we can build a diffusion convolutional layer that maps $P$-dimensional features to $Q$-dimensional outputs. Denote the parameter tensor as $\boldsymbol{\Theta} \in \mathbb{R}^{Q \times P \times K \times 2} = [\boldsymbol{\theta}]_{q,p}$, where $\boldsymbol{\Theta}_{q,p,:,:} \in \mathbb{R}^{K \times 2}$ parameterizes the convolutional filter for the $p$th input and the $q$th output. The diffusion convolutional layer is thus:

$$\boldsymbol{H}_{:,q} = \boldsymbol{a} \left( \sum_{p=1}^{P} \boldsymbol{X}_{:,p} \star_{\mathcal{G}} f_{\boldsymbol{\Theta}_{q,p,:,:}} \right) \qquad \text{for } q \in \{1, \cdots, Q\} \quad (3)$$

where $\boldsymbol{X} \in \mathbb{R}^{N \times P}$ is the input, $\boldsymbol{H} \in \mathbb{R}^{N \times Q}$ is the output, $\{f_{\boldsymbol{\Theta}_{q,p,:,:}}\}$ are the filters and $\boldsymbol{a}$ is the activation function (e.g., ReLU, Sigmoid). Diffusion convolutional layer learns the representations for graph structured data and we can train it using stochastic gradient based method.

**Relation with Spectral Graph Convolution**    Diffusion convolution is defined on both directed and undirected graphs. When applied to undirected graphs, we show that many existing graph structured convolutional operations including the popular spectral graph convolution, i.e., ChebNet (Defferrard et al., 2016), can be considered as a special case of diffusion convolution (up to a similarity transformation). Let $\boldsymbol{D}$ denote the degree matrix, and $\boldsymbol{L} = \boldsymbol{D}^{-\frac{1}{2}} (\boldsymbol{D} - \boldsymbol{W}) \boldsymbol{D}^{-\frac{1}{2}}$ be the normalized graph Laplacian, the following Proposition demonstrates the connection.

**Proposition 2.2.** *The spectral graph convolution defined as*

$$\boldsymbol{X}_{:,p} \star_{\mathcal{G}} f_{\boldsymbol{\theta}} = \boldsymbol{\Phi} \ F(\boldsymbol{\theta}) \ \boldsymbol{\Phi}^{\intercal} \boldsymbol{X}_{:,p}$$

*with eigenvalue decomposition $\boldsymbol{L} = \boldsymbol{\Phi} \boldsymbol{\Lambda} \boldsymbol{\Phi}^{\intercal}$ and $F(\boldsymbol{\theta}) = \sum_0^{K-1} \theta_k \boldsymbol{\Lambda}^k$, is equivalent to graph diffusion convolution up to a similarity transformation, when the graph $\mathcal{G}$ is undirected.*

*Proof.* See Appendix C.

## 2.3    TEMPORAL DYNAMICS MODELING

We leverage the recurrent neural networks (RNNs) to model the temporal dependency. In particular, we use Gated Recurrent Units (GRU) (Chung et al., 2014), which is a simple yet powerful variant of RNNs. We replace the matrix multiplications in GRU with the *diffusion convolution*, which leads to our proposed *Diffusion Convolutional Gated Recurrent Unit* (DCGRU).

$$\boldsymbol{r}^{(t)} = \sigma(\boldsymbol{\Theta}_r \star_{\mathcal{G}} [\boldsymbol{X}^{(t)}, \ \boldsymbol{H}^{(t-1)}] + \boldsymbol{b}_r) \qquad \boldsymbol{u}^{(t)} = \sigma(\boldsymbol{\Theta}_u \star_{\mathcal{G}} [\boldsymbol{X}^{(t)}, \ \boldsymbol{H}^{(t-1)}] + \boldsymbol{b}_u)$$
$$\boldsymbol{C}^{(t)} = \tanh(\boldsymbol{\Theta}_C \star_{\mathcal{G}} \left[ \boldsymbol{X}^{(t)}, \ (\boldsymbol{r}^{(t)} \odot \boldsymbol{H}^{(t-1)}) \right] + \boldsymbol{b}_c) \qquad \boldsymbol{H}^{(t)} = \boldsymbol{u}^{(t)} \odot \boldsymbol{H}^{(t-1)} + (1 - \boldsymbol{u}^{(t)}) \odot \boldsymbol{C}^{(t)}$$

where $\boldsymbol{X}^{(t)}, \boldsymbol{H}^{(t)}$ denote the input and output of at time $t$, $\boldsymbol{r}^{(t)}, \boldsymbol{u}^{(t)}$ are reset gate and update gate at time $t$, respectively. $\star_{\mathcal{G}}$ denotes the *diffusion convolution* defined in Equation 2 and $\boldsymbol{\Theta}_r, \boldsymbol{\Theta}_u, \boldsymbol{\Theta}_C$ are parameters for the corresponding filters. Similar to GRU, DCGRU can be used to build recurrent neural network layers and be trained using backpropagation through time.

In multiple step ahead forecasting, we employ the *Sequence to Sequence* architecture (Sutskever et al., 2014). Both the encoder and the decoder are recurrent neural networks with DCGRU. During training, we feed the historical time series into the encoder and use its final states to initialize the

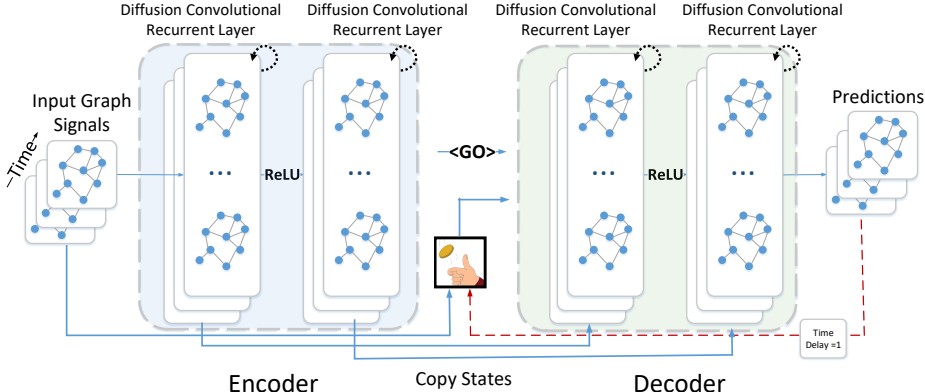

Figure 2: System architecture for the *Diffusion Convolutional Recurrent Neural Network* designed for spatiotemporal traffic forecasting. The historical time series are fed into an encoder whose final states are used to initialize the decoder. The decoder makes predictions based on either previous ground truth or the model output.

decoder. The decoder generates predictions given previous *ground truth observations*. At testing time, ground truth observations are replaced by predictions generated by the model itself. The discrepancy between the input distributions of training and testing can cause degraded performance. To mitigate this issue, we integrate *scheduled sampling* (Bengio et al., 2015) into the model, where we feed the model with either the ground truth observation with probability $\epsilon_i$ or the prediction by the model with probability $1 - \epsilon_i$ at the $i$th iteration. During the training process, $\epsilon_i$ gradually decreases to $0$ to allow the model to learn the testing distribution.

With both spatial and temporal modeling, we build a *Diffusion Convolutional Recurrent Neural Network* (DCRNN). The model architecture of DCRNN is shown in Figure 2. The entire network is trained by maximizing the likelihood of generating the target future time series using backpropagation through time. DCRNN is able to capture spatiotemporal dependencies among time series and can be applied to various spatiotemporal forecasting problems.

## 3 RELATED WORK

Traffic forecasting is a classic problem in transportation and operational research which are primarily based on queuing theory and simulations (Drew, 1968). Data-driven approaches for traffic forecasting have received considerable attention, and more details can be found in a recent survey paper (Vlahogianni et al., 2014) and the references therein. However, existing machine learning models either impose strong stationary assumptions on the data (e.g., auto-regressive model) or fail to account for highly non-linear temporal dependency (e.g., latent space model Yu et al. (2016); Deng et al. (2016)). Deep learning models deliver new promise for time series forecasting problem. For example, in Yu et al. (2017b); Laptev et al. (2017), the authors study time series forecasting using deep Recurrent Neural Networks (RNN). Convolutional Neural Networks (CNN) have also been applied to traffic forecasting. Zhang et al. (2016; 2017) convert the road network to a regular 2-D grid and apply traditional CNN to predict crowd flow. Cheng et al. (2017) propose DeepTransport which models the spatial dependency by explicitly collecting upstream and downstream neighborhood roads for each individual road and then conduct convolution on these neighborhoods respectively.

Recently, CNN has been generalized to arbitrary graphs based on the spectral graph theory. Graph convolutional neural networks (GCN) are first introduced in Bruna et al. (2014), which bridges the spectral graph theory and deep neural networks. Defferrard et al. (2016) propose ChebNet which improves GCN with fast localized convolutions filters. Kipf & Welling (2017) simplify ChebNet and achieve state-of-the-art performance in semi-supervised classification tasks. Seo et al. (2016) combine ChebNet with Recurrent Neural Networks (RNN) for structured sequence modeling. Yu et al. (2017a) model the sensor network as a undirected graph and applied ChebNet and convolutional sequence model (Gehring et al., 2017) to do forecasting. One limitation of the mentioned spectral based convolutions is that they generally require the graph to be undirected to calculate meaningful

Table 1: Performance comparison of different approaches for traffic speed forecasting. DCRNN achieves the best performance with all three metrics for all forecasting horizons, and the advantage becomes more evident with the increase of the forecasting horizon.

| | $T$ | Metric | HA | ARIMA$_{Kal}$ | VAR | SVR | FNN | FC-LSTM | *DCRNN* |
|---|---|---|---|---|---|---|---|---|---|
| METR-LA | 15 min | MAE | 4.16 | 3.99 | 4.42 | 3.99 | 3.99 | 3.44 | **2.77** |
| | | RMSE | 7.80 | 8.21 | 7.89 | 8.45 | 7.94 | 6.30 | **5.38** |
| | | MAPE | 13.0% | 9.6% | 10.2% | 9.3% | 9.9% | 9.6% | **7.3%** |
| | 30 min | MAE | 4.16 | 5.15 | 5.41 | 5.05 | 4.23 | 3.77 | **3.15** |
| | | RMSE | 7.80 | 10.45 | 9.13 | 10.87 | 8.17 | 7.23 | **6.45** |
| | | MAPE | 13.0% | 12.7% | 12.7% | 12.1% | 12.9% | 10.9% | **8.8%** |
| | 1 hour | MAE | 4.16 | 6.90 | 6.52 | 6.72 | 4.49 | 4.37 | **3.60** |
| | | RMSE | 7.80 | 13.23 | 10.11 | 13.76 | 8.69 | 8.69 | **7.59** |
| | | MAPE | 13.0% | 17.4% | 15.8% | 16.7% | 14.0% | 13.2% | **10.5%** |
| PEMS-BAY | 15 min | MAE | 2.88 | 1.62 | 1.74 | 1.85 | 2.20 | 2.05 | **1.38** |
| | | RMSE | 5.59 | 3.30 | 3.16 | 3.59 | 4.42 | 4.19 | **2.95** |
| | | MAPE | 6.8% | 3.5% | 3.6% | 3.8% | 5.19% | 4.8% | **2.9%** |
| | 30 min | MAE | 2.88 | 2.33 | 2.32 | 2.48 | 2.30 | 2.20 | **1.74** |
| | | RMSE | 5.59 | 4.76 | 4.25 | 5.18 | 4.63 | 4.55 | **3.97** |
| | | MAPE | 6.8% | 5.4% | 5.0% | 5.5% | 5.43% | 5.2% | **3.9%** |
| | 1 hour | MAE | 2.88 | 3.38 | 2.93 | 3.28 | 2.46 | 2.37 | **2.07** |
| | | RMSE | 5.59 | 6.50 | 5.44 | 7.08 | 4.98 | 4.96 | **4.74** |
| | | MAPE | 6.8% | 8.3% | 6.5% | 8.0% | 5.89% | 5.7% | **4.9%** |

spectral decomposition. Going from spectral domain to vertex domain, Atwood & Towsley (2016) propose diffusion-convolutional neural network (DCNN) which defines convolution as a diffusion process across each node in a graph-structured input. Hechtlinger et al. (2017) propose GraphCNN to generalize convolution to graph by convolving every node with its $p$ nearest neighbors. However, both these methods do not consider the temporal dynamics and mainly deal with static graph settings.

Our approach is different from all those methods due to both the problem settings and the formulation of the convolution on the graph. We model the sensor network as a weighted directed graph which is more realistic than grid or undirected graph. Besides, the proposed convolution is defined using bidirectional graph random walk and is further integrated with the sequence to sequence learning framework as well as the scheduled sampling to model the long-term temporal dependency.

## 4 EXPERIMENTS

We conduct experiments on two real-world large-scale datasets: (1) **METR-LA** This traffic dataset contains traffic information collected from loop detectors in the highway of Los Angeles County (Jagadish et al., 2014). We select 207 sensors and collect 4 months of data ranging from Mar 1st 2012 to Jun 30th 2012 for the experiment. (2) **PEMS-BAY** This traffic dataset is collected by California Transportation Agencies (CalTrans) Performance Measurement System (PeMS). We select 325 sensors in the Bay Area and collect 6 months of data ranging from Jan 1st 2017 to May 31th 2017 for the experiment. The sensor distributions of both datasets are visualized in Figure 8 in the Appendix.

In both of those datasets, we aggregate traffic speed readings into 5 minutes windows, and apply Z-Score normalization. 70% of data is used for training, 20% are used for testing while the remaining 10% for validation. To construct the sensor graph, we compute the pairwise road network distances between sensors and build the adjacency matrix using thresholded Gaussian kernel (Shuman et al., 2013). $W_{ij} = \exp\left(-\frac{\text{dist}(v_i, v_j)^2}{\sigma^2}\right)$ if $\text{dist}(v_i, v_j) \leq \kappa$, otherwise 0, where $W_{ij}$ represents the edge weight between sensor $v_i$ and sensor $v_j$, $\text{dist}(v_i, v_j)$ denotes the road network distance from sensor $v_i$ to sensor $v_j$. $\sigma$ is the standard deviation of distances and $\kappa$ is the threshold.

### 4.1 EXPERIMENTAL SETTINGS

**Baselines** We compare DCRNN[1] with widely used time series regression models, including (1) HA: Historical Average, which models the traffic flow as a seasonal process, and uses weighted

---

[1] The source code is available at https://github.com/liyaguang/DCRNN.

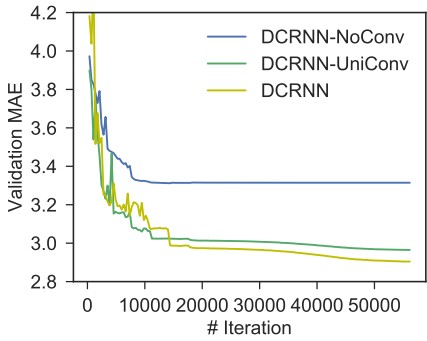
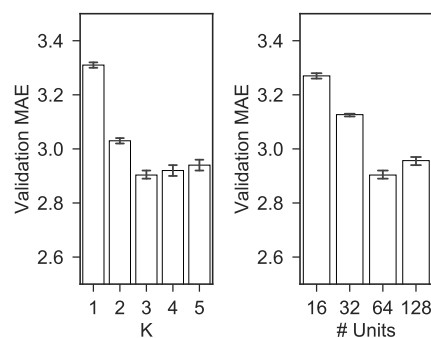

Figure 3: Learning curve for DCRNN and DCRNN without diffusion convolution. Removing diffusion convolution results in much higher validation error. Moreover, DCRNN with bidirectional random walk achieves the lowest validation error.

Figure 4: Effects of K and the number of units in each layer of DCRNN. K corresponds to the reception field width of the filter, and the number of units corresponds to the number of filters.

average of previous seasons as the prediction; (2) ARIMA$_{kal}$: Auto-Regressive Integrated Moving Average model with Kalman filter which is widely used in time series prediction; (3) VAR: Vector Auto-Regression (Hamilton, 1994). (4) SVR: Support Vector Regression which uses linear support vector machine for the regression task; The following deep neural network based approaches are also included: (5) Feed forward Neural network (FNN): Feed forward neural network with two hidden layers and L2 regularization. (6) Recurrent Neural Network with fully connected LSTM hidden units (FC-LSTM) (Sutskever et al., 2014).

All neural network based approaches are implemented using Tensorflow (Abadi et al., 2016), and trained using the Adam optimizer with learning rate annealing. The best hyperparameters are chosen using the Tree-structured Parzen Estimator (TPE) (Bergstra et al., 2011) on the validation dataset. Detailed parameter settings for DCRNN as well as baselines are available in Appendix E.

## 4.2 TRAFFIC FORECASTING PERFORMANCE COMPARISON

Table 1 shows the comparison of different approaches for 15 minutes, 30 minutes and 1 hour ahead forecasting on both datasets. These methods are evaluated based on three commonly used metrics in traffic forecasting, including (1) Mean Absolute Error (MAE), (2) Mean Absolute Percentage Error (MAPE), and (3) Root Mean Squared Error (RMSE). Missing values are excluded in calculating these metrics. Detailed formulations of these metrics are provided in Appendix E.2. We observe the following phenomenon in both of these datasets. (1) RNN-based methods, including FC-LSTM and DCRNN, generally outperform other baselines which emphasizes the importance of modeling the temporal dependency. (2) DCRNN achieves the best performance regarding all the metrics for all forecasting horizons, which suggests the effectiveness of spatiotemporal dependency modeling. (3) Deep neural network based methods including FNN, FC-LSTM and DCRNN, tend to have better performance than linear baselines for long-term forecasting, e.g., 1 hour ahead. This is because the temporal dependency becomes increasingly non-linear with the growth of the horizon. Besides, as the historical average method does not depend on short-term data, its performance is invariant to the small increases in the forecasting horizon.

Note that, traffic forecasting on the METR-LA (Los Angeles, which is known for its complicated traffic conditions) dataset is more challenging than that in the PEMS-BAY (Bay Area) dataset. Thus we use METR-LA as the default dataset for following experiments.

## 4.3 EFFECT OF SPATIAL DEPENDENCY MODELING

To further investigate the effect of spatial dependency modeling, we compare DCRNN with the following variants: (1) DCRNN-NoConv, which ignores spatial dependency by replacing the transition matrices in the diffusion convolution (Equation 2) with identity matrices. This essentially means the forecasting of a sensor can be only be inferred from its own historical readings; (2) DCRNN-UniConv,

Table 2: Performance comparison for DCRNN and GCRNN on the METRA-LA dataset.

|  | 15 min | | | 30 min | | | 1 hour | | |
|---|---|---|---|---|---|---|---|---|---|
|  | MAE | RMSE | MAPE | MAE | RMSE | MAPE | MAE | RMSE | MAPE |
| DCRNN | **2.77** | **5.38** | **7.3%** | **3.15** | **6.45** | **8.8%** | **3.60** | **7.60** | **10.5%** |
| GCRNN | 2.80 | 5.51 | 7.5% | 3.24 | 6.74 | 9.0% | 3.81 | 8.16 | 10.9% |

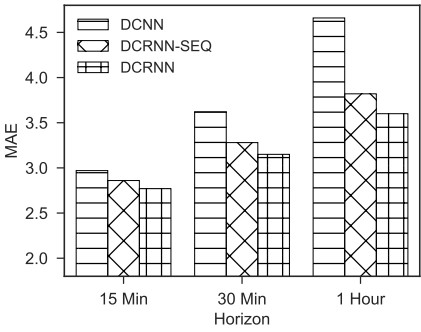

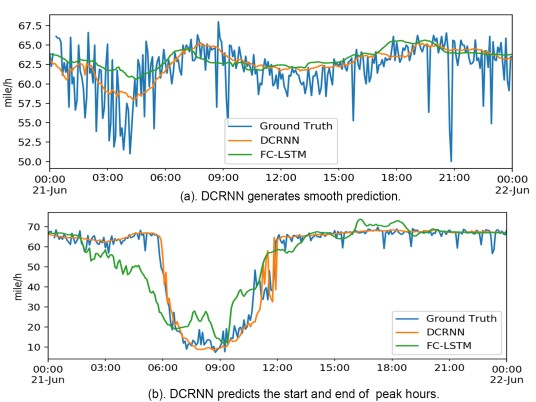

Figure 6: Traffic time series forecasting visualization. DCRNN generates smooth prediction and is usually better at predict the start and end of peak hours.

Figure 5: Performance comparison for different DCRNN variants. DCRNN, with the sequence to sequence framework and scheduled sampling, achieves the lowest MAE on the validation dataset. The advantage becomes more clear with the increase of the forecasting horizon.

which only uses the forward random walk transition matrix for diffusion convolution; Figure 3 shows the learning curves of these three models with roughly the same number of parameters. Without diffusion convolution, DCRNN-NoConv has much higher validation error. Moreover, DCRNN achieves the lowest validation error which shows the effectiveness of using bidirectional random walk. The intuition is that the bidirectional random walk gives the model the ability and flexibility to capture the influence from both the upstream and the downstream traffic.

To investigate the effect of graph construction, we construct a undirected graph by setting $\widehat{W}_{ij} = \widehat{W}_{ji} = \max(W_{ij}, W_{ji})$, where $\widehat{W}$ is the new symmetric weight matrix. Then we develop a variant of DCRNN denotes GCRNN, which uses the sequence to sequence learning with *ChebNet graph convolution* (Equation 5) with roughly the same amount of parameters. Table 2 shows the comparison between DCRNN and GCRNN in the METR-LA dataset. DCRNN consistently outperforms GCRNN. The intuition is that directed graph better captures the asymmetric correlation between traffic sensors. Figure 4 shows the effects of different parameters. $K$ roughly corresponds to the size of filters' reception fields while the number of units corresponds to the number of filters. Larger $K$ enables the model to capture broader spatial dependency at the cost of increasing learning complexity. We observe that with the increase of $K$, the error on the validation dataset first quickly decrease, and then slightly increase. Similar behavior is observed for varying the number of units.

## 4.4 EFFECT OF TEMPORAL DEPENDENCY MODELING

To evaluate the effect of temporal modeling including the sequence to sequence framework as well as the scheduled sampling mechanism, we further design three variants of DCRNN: (1) DCNN: in which we concatenate the historical observations as a fixed length vector and feed it into stacked diffusion convolutional layers to predict the future time series. We train a single model for one step ahead prediction, and feed the previous prediction into the model as input to perform multiple steps ahead prediction. (2) DCRNN-SEQ: which uses the encoder-decoder sequence to sequence learning framework to perform multiple steps ahead forecasting. (3) DCRNN: similar to DCRNN-SEQ except for adding scheduled sampling.

Figure 7: Visualization of learned localized filters centered at different nodes with $K = 3$ on the METR-LA dataset. The star denotes the center, and the colors represent the weights. We observe that weights are localized around the center, and diffuse alongside the road network.

Figure 5 shows the comparison of those four methods with regards to MAE for different forecasting horizons. We observe that: (1) DCRNN-SEQ outperforms DCNN by a large margin which conforms the importance of modeling temporal dependency. (2) DCRNN achieves the best result, and its superiority becomes more evident with the increase of the forecasting horizon. This is mainly because the model is trained to deal with its mistakes during multiple steps ahead prediction and thus suffers less from the problem of error propagation. We also train a model that always been fed its output as input for multiple steps ahead prediction. However, its performance is much worse than all the three variants which emphasizes the importance of scheduled sampling.

## 4.5  MODEL INTERPRETATION

To better understand the model, we visualize forecasting results as well as learned filters. Figure 6 shows the visualization of 1 hour ahead forecasting. We have the following observations: (1) DCRNN generates smooth prediction of the mean when small oscillation exists in the traffic speeds (Figure 6(a)). This reflects the robustness of the model. (2) DCRNN is more likely to accurately predict abrupt changes in the traffic speed than baseline methods (e.g., FC-LSTM). As shown in Figure 6(b), DCRNN predicts the start and the end of the peak hours. This is because DCRNN captures the spatial dependency, and is able to utilize the speed changes in neighborhood sensors for more accurate forecasting. Figure 7 visualizes examples of learned filters centered at different nodes. The star denotes the center, and colors denote the weights. We can observe that (1) weights are well localized around the center, and (2) the weights diffuse based on road network distance. More visualizations are provided in Appendix F.

## 5  CONCLUSION

In this paper, we formulated the traffic prediction on road network as a spatiotemporal forecasting problem, and proposed the *diffusion convolutional recurrent neural network* that captures the spatiotemporal dependencies. Specifically, we use bidirectional graph random walk to model spatial dependency and recurrent neural network to capture the temporal dynamics. We further integrated the encoder-decoder architecture and the scheduled sampling technique to improve the performance for long-term forecasting. When evaluated on two large-scale real-world traffic datasets, our approach obtained significantly better prediction than baselines. For future work, we will investigate the following two aspects (1) applying the proposed model to other spatial-temporal forecasting tasks; (2) modeling the spatiotemporal dependency when the underlying graph structure is evolving, e.g., the K nearest neighbor graph for moving objects.

ACKNOWLEDGMENTS

This research has been funded in part by NSF grants CNS-1461963, IIS-1254206, IIS-1539608, Caltrans-65A0533, the USC Integrated Media Systems Center (IMSC), and the USC METRANS Transportation Center. Any opinions, findings, and conclusions or recommendations expressed in this material are those of the authors and do not necessarily reflect the views of any of the sponsors such as NSF. Also, the authors would like to thank Shang-Hua Teng, Dehua Cheng and Siyang Li for helpful discussions and comments.

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

APPENDIX

A   NOTATION

Table 3: Notation

| Name | |
|---|---|
| $\mathcal{G}$ | a graph |
| $\mathcal{V}, v_i$ | nodes of a graph, $|\mathcal{V}| = N$ and the $i$-th node. |
| $\mathcal{E}$ | edges of a graph |
| $\boldsymbol{W}, W_{ij},$ | weight matrix of a graph and its entries |
| $\boldsymbol{D}, \boldsymbol{D}_I, \boldsymbol{D}_O$ | undirected degree matrix, In-degree/out-degree matrix |
| $\boldsymbol{L}$ | normalized graph Laplacian |
| $\boldsymbol{\Phi}, \boldsymbol{\Lambda}$ | eigen-vector matrix and eigen-value matrix of $\boldsymbol{L}$ |
| $\boldsymbol{X}, \hat{\boldsymbol{X}} \in \mathbb{R}^{N \times P}$ | a graph signal, and the predicted graph signal. |
| $\boldsymbol{X}^{(t)} \in \mathbb{R}^{N \times P}$ | a graph signal at time $t$. |
| $\boldsymbol{H} \in \mathbb{R}^{N \times Q}$ | output of the diffusion convolutional layer. |
| $f_{\boldsymbol{\theta}}, \boldsymbol{\theta}$ | convolutional filter and its parameters. |
| $f_{\boldsymbol{\Theta}}, \boldsymbol{\Theta}$ | convolutional layer and its parameters. |

Table 3 summarizes the main notations used in the paper.

B   EFFICIENT CALCULATION OF EQUATION 2

Equation 2 can be decomposed into two parts with the same time complexity, i.e., one part with $\boldsymbol{D}_O^{-1}\boldsymbol{W}$ and the other part with $\boldsymbol{D}_I^{-1}\boldsymbol{W}^\mathsf{T}$. Thus we will only show the time complexity of the first part.

Let $T_k(\boldsymbol{x}) = \left(\boldsymbol{D}_O^{-1}\boldsymbol{W}\right)^k \boldsymbol{x}$, The first part of Equation 2 can be rewritten as

$$\sum_{k=0}^{K-1} \theta_k T_k(X_{:,p}) \tag{4}$$

As $T_{k+1}(\boldsymbol{x}) = \boldsymbol{D}_O^{-1}\boldsymbol{W} T_k(\boldsymbol{x})$ and $\boldsymbol{D}_O^{-1}\boldsymbol{W}$ is sparse, it is easy to see that Equation 4 can be calculated using $O(K)$ recursive sparse-dense matrix multiplication each with time complexity $O(|\mathcal{E}|)$. Consequently, the time complexities of both Equation 2 and Equation 4 are $O(K|\mathcal{E}|)$. For dense graph, we may use spectral sparsification (Cheng et al., 2015) to make it sparse.

C   RELATION WITH SPECTRAL GRAPH CONVOLUTION

*Proof.* The spectral graph convolution utilizes the concept of normalized graph Laplacian $\boldsymbol{L} = \boldsymbol{D}^{-\frac{1}{2}}(\boldsymbol{D} - \boldsymbol{W})\boldsymbol{D}^{-\frac{1}{2}} = \boldsymbol{\Phi}\boldsymbol{\Lambda}\boldsymbol{\Phi}^\mathsf{T}$. ChebNet parametrizes $f_\theta$ to be a $K$ order polynomial of $\boldsymbol{\Lambda}$, and calculates it using stable Chebyshev polynomial basis.

$$\boldsymbol{X}_{:,p} \star_{\mathcal{G}} f_{\boldsymbol{\theta}} = \boldsymbol{\Phi} \left( \sum_{k=0}^{K-1} \theta_k \boldsymbol{\Lambda}^k \right) \boldsymbol{\Phi}^\mathsf{T} \boldsymbol{X}_{:,p} = \sum_{k=0}^{K-1} \theta_k \boldsymbol{L}^k \boldsymbol{X}_{:,p} = \sum_{k=0}^{K-1} \tilde{\theta}_k T_k(\tilde{\boldsymbol{L}}) \boldsymbol{X}_{:,p} \tag{5}$$

where $T_0(x) = 1, T_1(x) = x, T_k(x) = xT_{k-1}(x) - T_{k-2}(x)$ are the basis of the Cheyshev polynomial. Let $\lambda_{max}$ denote the largest eigenvalue of $\boldsymbol{L}$, and $\tilde{\boldsymbol{L}} = \frac{2}{\lambda_{max}}\boldsymbol{L} - \boldsymbol{I}$ represents a rescaling of the graph Laplacian that maps the eigenvalues from $[0, \lambda_{max}]$ to $[-1, 1]$ since Chebyshev polynomial forms an orthogonal basis in $[-1, 1]$. Equation 5 can be considered as a polynomial of $\tilde{\boldsymbol{L}}$ and we will show that the output of ChebNet Convolution is *similar* to the output of diffusion convolution up to constant scaling factor. Assume $\lambda_{max} = 2$ and $\boldsymbol{D}_I = \boldsymbol{D}_O = \boldsymbol{D}$ for undirected graph.

$$\tilde{\boldsymbol{L}} = \boldsymbol{D}^{-\frac{1}{2}}(\boldsymbol{D} - \boldsymbol{W})\boldsymbol{D}^{-\frac{1}{2}} - \boldsymbol{I} = -\boldsymbol{D}^{-\frac{1}{2}}\boldsymbol{W}\boldsymbol{D}^{-\frac{1}{2}} \sim -\boldsymbol{D}^{-1}\boldsymbol{W} \tag{6}$$

$\tilde{\boldsymbol{L}}$ is *similar* to the negative random walk transition matrix, thus the output of Equation 5 is also similar to the output of Equation 2 up to constant scaling factor. $\qquad\square$

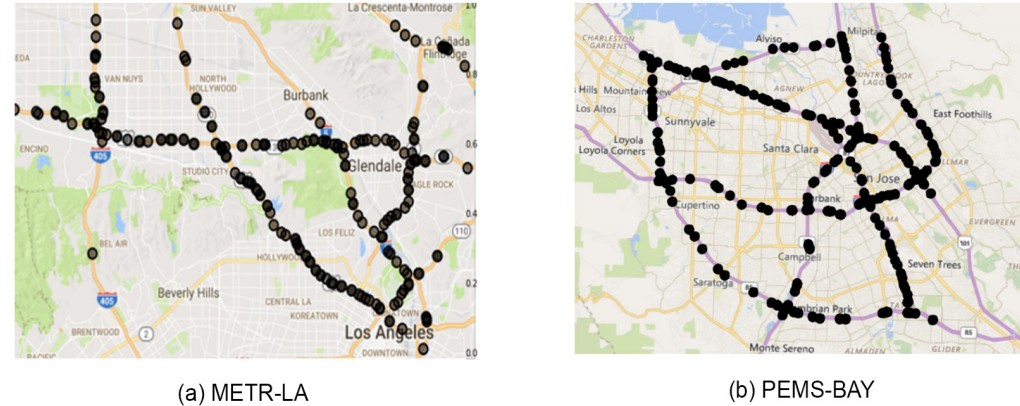

(a) METR-LA                                    (b) PEMS-BAY

Figure 8: Sensor distribution of the METR-LA and PEMS-BAY dataset.

## D   MORE RELATED WORK AND DISCUSSION

Xie et al. (2010) introduce a Gaussian processes (GPs) based method. GPs are hard to scale to the large dataset and are generally not suitable for relatively long-term traffic prediction like 1 hour (i.e.,12 steps ahead), as the variance can be accumulated and becomes extremely large.

Cai et al. (2016) propose to use spatiotemporal nearest neighbor for traffic forecasting (ST-KNN). Though ST-KNN considers both the spatial and the temporal dependencies, it has the following drawbacks. As shown in Fusco et al. (2016), ST-KNN performs independent forecasting for each individual road. The prediction of a road is a weighted combination of its own historical traffic speeds. This makes it hard for ST-KNN to fully utilize information from neighbors. Besides, ST-KNN is a non-parametric approach and each road is modeled and calculated separately (Cai et al., 2016), which makes it hard to generalize to unseen situations and to scale to large datasets. Finally, in ST-KNN, all the similarities are calculated using hand-designed metrics with few learnable parameters, and this may limit its representational power.

Cheng et al. (2017) propose DeepTransport which models the spatial dependency by explicitly collecting certain number of upstream and downstream roads for each individual road and then conduct convolution on these roads respectively. Comparing with Cheng et al. (2017), DCRNN models the spatial dependency in a more systematic way, i.e., generalizing convolution to the traffic sensor graph based on the diffusion nature of traffic. Besides, we derive DCRNN from the property of random walk and show that the popular spectral convolution ChebNet is a special case of our method.

The proposed approach is also related to graph embedding techniques, e.g., Deepwalk (Perozzi et al., 2014), node2vec (Grover & Leskovec, 2016) which learn a low dimension representation for each node in the graph. DCRNN also learns a representation for each node. The learned representations capture both the spatial and the temporal dependency and at the same time are optimized with regarding to the objective, e.g., future traffic speeds.

## E   DETAILED EXPERIMENTAL SETTINGS

**HA**   Historical Average, which models the traffic flow as a seasonal process, and uses weighted average of previous seasons as the prediction. The period used is 1 week, and the prediction is based on aggregated data from previous weeks. For example, the prediction for this Wednesday is the averaged traffic speeds from last four Wednesdays. As the historical average method does not depend on short-term data, its performance is invariant to the small increases in the forecasting horizon

**ARIMA**$_{kal}$   : Auto-Regressive Integrated Moving Average model with Kalman filter. The orders are (3, 0, 1), and the model is implemented using the *statsmodel* python package.

**VAR**   Vector Auto-regressive model (Hamilton, 1994). The number of lags is set to 3, and the model is implemented using the *statsmodel* python package.

**SVR**   Linear Support Vector Regression, the penalty term $C = 0.1$, the number of historical observation is 5.

The following deep neural network based approaches are also included.

**FNN**   Feed forward neural network with two hidden layers, each layer contains 256 units. The initial learning rate is $1e^{-3}$, and reduces to $\frac{1}{10}$ every 20 epochs starting at the 50th epochs. In addition, for all hidden layers, dropout with ratio 0.5 and L2 weight decay $1e^{-2}$ is used. The model is trained with batch size 64 and MAE as the loss function. Early stop is performed by monitoring the validation error.

**FC-LSTM**   The Encoder-decoder framework using LSTM with peephole (Sutskever et al., 2014). Both the encoder and the decoder contain two recurrent layers. In each recurrent layer, there are 256 LSTM units, L1 weight decay is $2e^{-5}$, L2 weight decay $5e^{-4}$. The model is trained with batch size 64 and loss function MAE. The initial learning rate is 1e-4 and reduces to $\frac{1}{10}$ every 10 epochs starting from the 20th epochs. Early stop is performed by monitoring the validation error.

**DCRNN**   : Diffusion Convolutional Recurrent Neural Network. Both encoder and decoder contain two recurrent layers. In each recurrent layer, there are 64 units, the initial learning rate is $1e^{-2}$, and reduces to $\frac{1}{10}$ every 10 epochs starting at the 20th epoch and early stopping on the validation dataset is used. Besides, the maximum steps of random walks, i.e., $K$, is set to 3. For scheduled sampling, the thresholded inverse sigmoid function is used as the probability decay:

$$\epsilon_i = \frac{\tau}{\tau + \exp\left(i/\tau\right)}$$

where $i$ is the number of iterations while $\tau$ are parameters to control the speed of convergence. $\tau$ is set to 3,000 in the experiments. The implementation is available in `https://github.com/liyaguang/DCRNN`.

### E.1   DATASET

We conduct experiments on two real-world large-scale datasets:

- **METR-LA** This traffic dataset contains traffic information collected from loop detectors in the highway of Los Angeles County (Jagadish et al., 2014). We select 207 sensors and collect 4 months of data ranging from Mar 1st 2012 to Jun 30th 2012 for the experiment. The total number of observed traffic data points is 6,519,002.

- **PEMS-BAY** This traffic dataset is collected by California Transportation Agencies (CalTrans) Performance Measurement System (PeMS). We select 325 sensors in the Bay Area and collect 6 months of data ranging from Jan 1st 2017 to May 31th 2017 for the experiment. The total number of observed traffic data points is 16,937,179.

The sensor distributions of both datasets are visualized in Figure 8.

In both of those datasets, we aggregate traffic speed readings into 5 minutes windows, and apply Z-Score normalization. 70% of data is used for training, 20% are used for testing while the remaining 10% for validation. To construct the sensor graph, we compute the pairwise road network distances between sensors and build the adjacency matrix using thresholded Gaussian kernel (Shuman et al., 2013).

$$W_{ij} = \exp\left(-\frac{\text{dist}(v_i, v_j)^2}{\sigma^2}\right) \quad \text{if } \text{dist}(v_i, v_j) \leq \kappa, \text{ otherwise } 0$$

where $W_{ij}$ represents the edge weight between sensor $v_i$ and sensor $v_j$, $\text{dist}(v_i, v_j)$ denotes the road network distance from sensor $v_i$ to sensor $v_j$. $\sigma$ is the standard deviation of distances and $\kappa$ is the threshold.

### E.2 METRICS

Suppose $\boldsymbol{x} = x_1, \cdots, x_n$ represents the ground truth, $\hat{\boldsymbol{x}} = \hat{x}_1, \cdots, \hat{x}_n$ represents the predicted values, and $\Omega$ denotes the indices of observed samples, the metrics are defined as follows.

Root Mean Square Error (RMSE)

$$\text{RMSE}(\boldsymbol{x}, \hat{\boldsymbol{x}}) = \sqrt{\frac{1}{|\boldsymbol{\Omega}|} \sum_{i \in \boldsymbol{\Omega}} (x_i - \hat{x}_i)^2}$$

Mean Absolute Percentage Error (MAPE)

$$\text{MAPE}(\boldsymbol{x}, \hat{\boldsymbol{x}}) = \frac{1}{|\boldsymbol{\Omega}|} \sum_{i \in \boldsymbol{\Omega}} \left| \frac{x_i - \hat{x}_i}{x_i} \right|$$

Mean Absolute Error (MAE)

$$\text{MAE}(\boldsymbol{x}, \hat{\boldsymbol{x}}) = \frac{1}{|\boldsymbol{\Omega}|} \sum_{i \in \boldsymbol{\Omega}} |x_i - \hat{x}_i|$$

### F MODEL VISUALIZATION

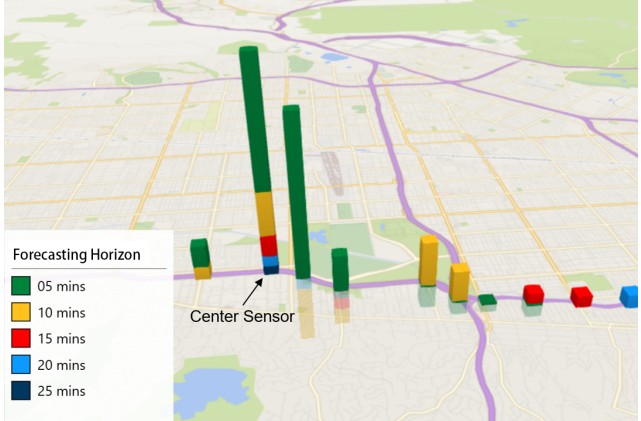

Figure 9: Sensor correlations between the center sensor and its neighborhoods for different forecasting horizons. The correlations are estimated using regularized VAR. We observe that the correlations are localized and closer neighborhoods usually have larger relevance, and the magnitude of correlation quickly decay with the increase of distance which is consistent with the diffusion process on the graph.

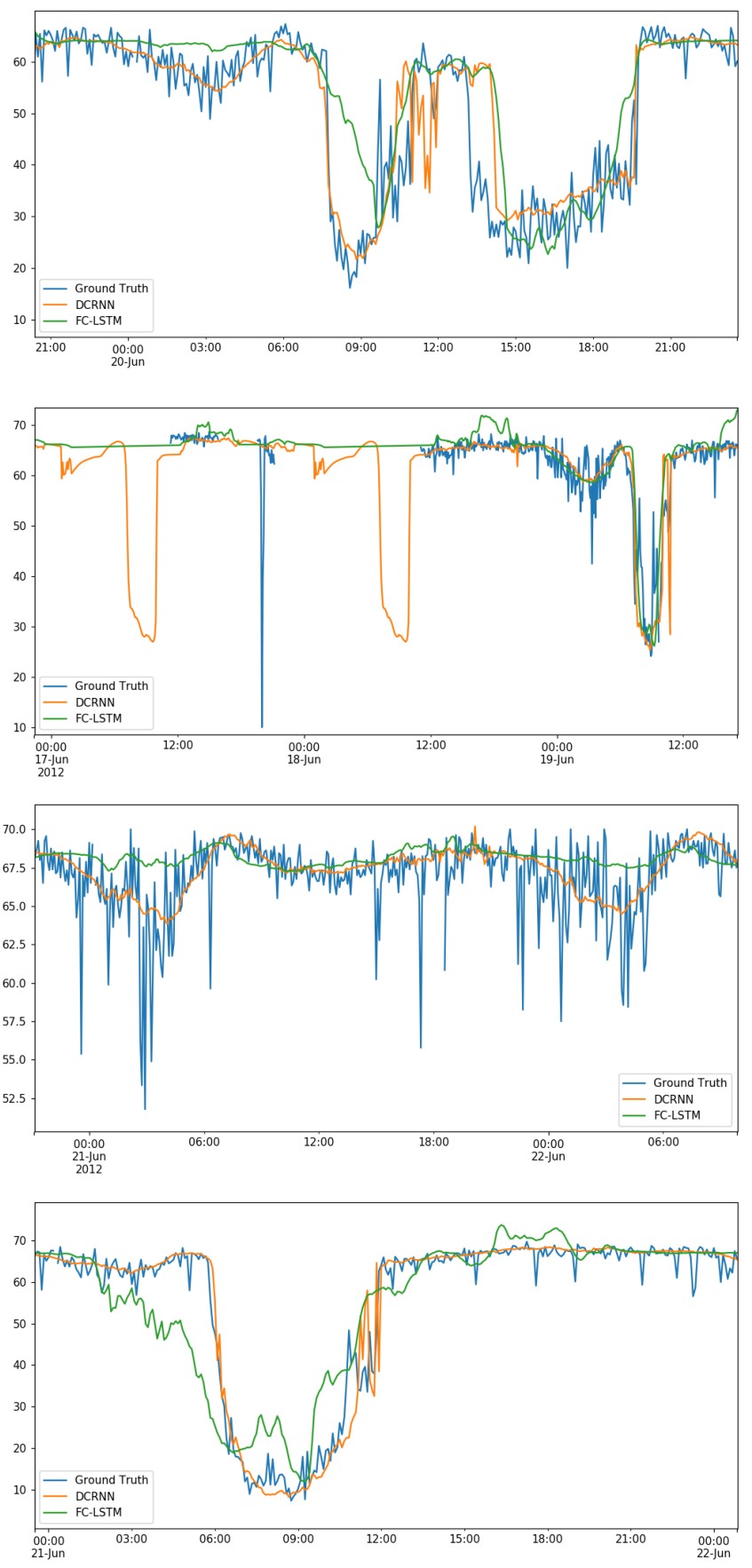

Figure 10: Traffic time series forecasting visualization.

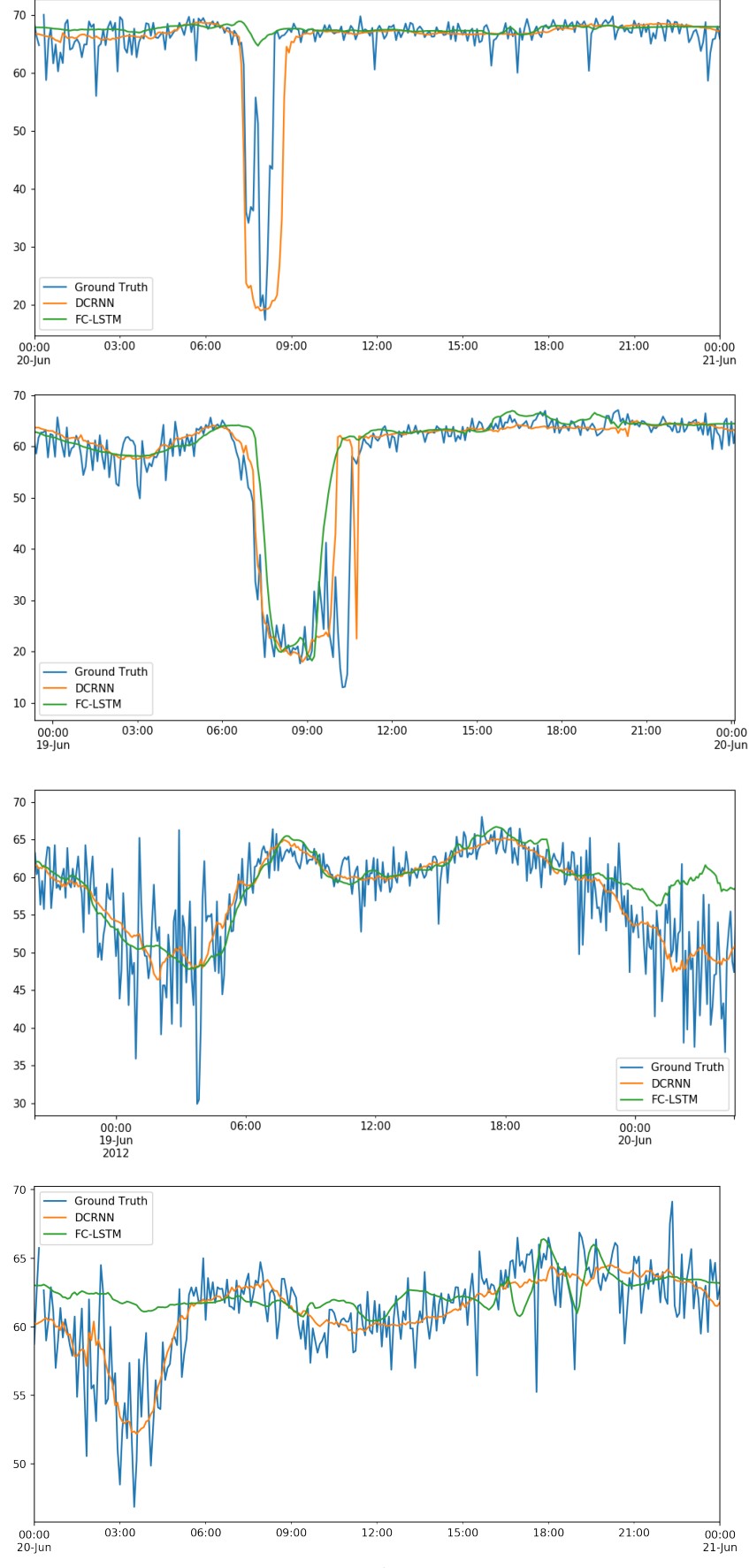

Figure 11: Traffic time series forecasting visualization.