# OpenReview forum: "Diffusion Convolutional Recurrent Neural Network: Data-Driven Traffic Forecasting"
_ICLR.cc/2018/Conference — Accept (Poster)_

### Official Review · AnonReviewer1 · 2017-11-27
**The paper proposes the Diffusion Convolutional Recurrent Neural Network architecture for the spatiotemporal traffic forecasting problem. Overall, the paper is well written with incremental technical contributions.**

**Rating:** 5
**Confidence:** 3

**Review:**

The paper proposes to build a graph where the edge weight is defined using the road network distance which is shown to be more realistic than the Euclidean distance. The defined diffusion convolution operation is essentially conducting random walks over the road segment graph. To avoid the expensive matrix operation for the random walk, it empirically shows that K = 3 hops of the random walk can give a good performance. The outputs of the graph convolutionary operation are then fed into the sequence to sequence architecture with the GRU cell to model the temporal dependency. Experiments show that the proposed architecture can achieve good performance compared to classic time series baselines and several simplified variants of the proposed model.

Although the paper argues that several existing deep-learning based approaches may not be directly applied in the current setting either due to using Euclidean distance or undirected graph structure, the comparisons are not persuasive. For example, the approach in the paper "DeepTransport: Learning Spatial-Temporal Dependency for Traffic Condition Forecasting" also consider directed graph and a diffusion effect from 2 or 3 hops away in the neighboring subgraph of a target road segment.

Furthermore, the paper proposes to use two convolution components in Equation 2, each of which corresponds to out-degree and in-degree direction, respectively. This effectively increase the number of model parameters to learn. Compared to the existing spectral graph convolution approach, it is still not clear how its performance will be by using the same number of parameters. The experiments will be improved if it can compare with "Spatio-temporal graph convolutional neural network: A deep learning framework for traffic forecasting" using roughly the same number of parameters.

---

> ### Author Response · Authors · 2017-12-28
> **Response to ICLR 2018 Conference Paper804 AnonReviewer1**
>
> We sincerely thank the reviewer for the detailed and helpful comments. First, we would like to clarify the main contribution of our work: the proposed model, DCRNN, achieves significant performance gain with theoretical justifications -  we derive diffusion convolution from the property of random walk [1] capturing the diffusion nature of traffic. And we show in theory that the popular spectral convolution, i.e., ChebNet [2], is a special case of our method (Proposition 2.2). We further incorporate diffusion convolution with RNN in a non-trivial way and improve long-term forecasting performance with scheduled sampling. Compared with existing work, our model is one of the first with significant performance improvement in traffic prediction and insightful theoretical justifications.
>
> Below we address specific questions/concerns:
> >> "Compared to the existing spectral graph convolution approach, it is still not clear how its performance will be by using the same number of parameters."
> In Table 2, we compare to ChebNet [4], one of the most popular spectral graph convolutions with roughly the same number of parameters, and our approach achieves improved results.
>
> Here is the detailed architecture of these models:
> DCRNN (proposed model),  K=3, #layers=2,  # filters in each layer=64, #params=373,376
> GCRNN (ChebNet + GRU), K=3, #layers=2,  # filters in each layer=83, #params=376,156
>
> >> Comparison with DeepTransport [3].
>
> DeepTransport [3] is a very recent unpublished work and has not been peer-reviewed, it first becomes available on arXiv on Sep 27th, 2017. In [3], the authors model the spatial dependency by explicitly collecting upstream and downstream roads for each individual road and then conduct traditional convolution on these roads respectively. Comparing with [3], we model the spatial dependency in a more systematic way, i.e., generalizing convolution to the traffic sensor graph based on the diffusion nature of traffic. Besides, we derive DCRNN from the property of random walk and show that the popular spectral convolution ChebNet is a special case of our method.
>
> We will reference and differentiate those work in the next version.
>
> Reference:
> [1] S. Teng et al. Scalable algorithms for data and network analysis. Foundations and Trends
> R in Theoretical Computer Science.
> [2] M.l Defferrard et al. Convolutional Neural Networks on Graphs with Fast Localized Spectral Filtering.
> [3] X. Cheng et al. DeepTransport: Learning Spatial-Temporal Dependency for Traffic Condition Forecasting. http://arxiv.org/abs/1709.09585

---

### Official Review · AnonReviewer3 · 2017-11-27
**The paper presents an interesting approach to a relevant topic but lacks a comparison to other models that include spatio-temporal dependencies for traffic forecasting.**

**Rating:** 4
**Confidence:** 5

**Review:**

The paper adresses an important task to build a data-driven model traffic forecasting model. The paper takes into consideration the spatio-temporal autocorralation and tackles this with a diffusion process for convolutional recurrent neural networks. The paper lacks a comparison to other models that aim to include spatio-temporal dependencies used for this problem - namely Gaussian Processes, spatial k-NN.
The paper motivates the goal to obtain smooth traffic predictions, but traffic is not a smooth process, e.g. traffic lights and intersections cause non smooth effects. therefore it is difficult to follow this argumentation. Statements as 'is usually better at predicting start and end of peak hours' (caption of Figure 6) should be supported by statistical test that stress significance of the statement.
The method performs well on the presented data - in comparison to the models that do not consider autocorrelation. This might be because tests were not performed with commonly used traffic models or the traffic data was in favour for this model - it remains unclear, whether the proposed method really contributes better predictions or higher scalibility or faster computation to the mentioned problem. How does the proposed model behave in case of a shift in the traffic distribution? How do sudden changes (accident, road closure, etc.) affect the performance?

---

> ### Author Response · Authors · 2017-12-28
> **Response to ICLR 2018 Conference Paper804 AnonReviewer3**
>
> Response to ICLR 2018 Conference Paper804 AnonReviewer3
> We sincerely thank the reviewer for the detailed comments. However, we cannot agree with the reviewer’s assessment regarding the significance of this paper. The paper proposes a novel Diffusion Convolutional Recurrent Neural Network which captures both the spatial and temporal dependencies among traffic time series with theoretical justifications. We derive diffusion convolution from the property of random walk [1] capturing the diffusion nature of traffic, and we show in theory that the popular spectral convolution, i.e., ChebNet [2], is a special case of our method (Proposition 2.2). We further incorporate diffusion convolution with RNN in a non-trivial way and improve long-term forecasting performance with scheduled sampling. Compared with existing work, our model is one of the first with significant performance improvement in traffic prediction and insightful theoretical justifications.
>
> Below we address specific questions/concerns:
>
> >> "The paper lacks a comparison to other models that aim to include spatiotemporal dependencies used for this problem - namely Gaussian Processes, spatial k-NN."
>
> We do compare models that consider spatial and temporal dependencies, e.g., VAR and FC-LSTM, which take as input time series from all sensors, and model both the dependency among different sensors and the dependency among different time steps.
>
> Comparison with Gaussian Processes (GPs) [3] and Spatial KNN [4]:
> - GPs are hard to scale to the large dataset and are generally not suitable for relatively long-term traffic prediction like 1 hour (i.e.,12 steps ahead), as the variance can be accumulated and becomes extremely large.
> - Though spatial KNN [4] considers both the spatial and the temporal dependencies, it has the following drawbacks:
>     1) As shown in [5], Spatial KNN performs independent forecasting for each individual road. The prediction of a road is a weighted combination of its own historical traffic speeds. This makes it hard for Spatial KNN to fully utilize information from neighbors.
>     2) Spatial KNN is a non-parametric approach and each road is modeled and calculated separately [5], which may make it hard to generalize to unseen situations and to scale to large datasets.
>    3) In Spatial KNN, all the similarities are calculated using hand-designed metrics with few learnable parameters which may limit its representation power.
>
> We will add more discussion about these methods in the next version.
>
> >> "The paper motivates the goal to obtain smooth traffic predictions..."
> Our motivation is to capture the temporal and spatial dependency among traffic time series rather than to generate smooth predictions. The reviewer may misunderstand the description of Fig. 6. Though in Fig. 6, we show that the model's predictions are usually smooth, this is not our motivation. The smoothness can be explained by the following facts:
> - The objective is to predict the traffic speed averaged over 5 mins which is usually smooth even with traffic lights and intersections.
> - The traffic speed time series are collected from the highway where traffic lights and intersections are less common.
>
> Besides, for clarification, by saying "traffic signal" we mean the graph signal of traffic as described in Section 2.1 rather than "traffic light".
>
> >> "The method performs well on the presented data - in comparison to the models that do not consider autocorrelation"
> In fact, most baselines that our method beat do consider autocorrelation. For example, VAR, FC-LSTM, which take as input multiple time series, are able to model the autocorrelation among different time series and among different time steps. Specifically, we visualize the spatial-temporal dependencies captured by the VAR model in Fig. 9.
>
> >>  "This might be because tests were not performed with commonly used traffic models or the traffic data was in favour for this model"
> We do not agree with this comment. We validate our methods on two real-world datasets and have observed consistent improvements. We compared the proposed model with ARIMA, historical average, SVR, VAR, FNN and LSTM based method, which are widely used for traffic forecasting [6, 7, 8].
> Besides, the datasets we used come from Los Angeles and the Bay Area/San Francisco, with different traffic patterns. These datasets are fair representatives for traffic prediction. In addition, we want to emphasize that the traffic sensors were not cherry-picked in our experiments. We simply chose an area and performed experiments on all the working sensors in that area.

---

> > ### Author Response · Authors · 2017-12-28
> > **Response to ICLR 2018 Conference Paper804 AnonReviewer3**
> >
> > >> "How does the proposed model behave in case of a shift in the traffic distribution? How do sudden changes (accident, road closure, etc.) affect the performance?"
> > We have not explicitly studied the model's behavior in case of the shift in the traffic distribution or accidents. Generally, RNNs can capture nonlinear dynamics better than traditional linear models [9] and the diffusion convolution can model the dependency among nearby sensors which should be helpful. Finally, studying the effect of sudden changes is an interesting problem and can be a potential future work.
> >
> > Reference:
> > [1] S. Teng et al. Scalable algorithms for data and network analysis. Foundations and Trends
> > R in Theoretical Computer Science.
> > [2] M.l Defferrard et al. Convolutional Neural Networks on Graphs with Fast Localized Spectral Filtering
> > [3] Y. Xie et al. Gaussian Processes for Short-Term Traffic Volume Forecasting.
> > [4] P. Cai  et al. A spatiotemporal correlative k-nearest neighbor model for short-term traffic multi-step forecasting
> > [5] G. Fusco. Short-term speed predictions exploiting big data on large urban road networks.
> > [6] S. R. Chandra and H. Al-Deek. Predictions of Freeway Traffic Speeds and Volumes Using Vector Autoregressive Models.
> > [7] M Lippi et al. Short-Term Trafﬁc Flow Forecasting: An Experimental Comparison of Time-Series Analysis and Supervised Learning.
> > [8] B. L. Smith et al. Comparison of parametric and nonparametric models for traﬃc ﬂow forecasting.
> > [9] J. T. Connor et al. Recurrent Neural Networks and Robust Time Series Prediction.

---

### Official Review · AnonReviewer2 · 2017-11-28
**This work proposed a Diffusion Convolutional Recurrent Neural Network framework for traffic forecasting that incorporates both spatial and temporal dependency in the traffic flow. Specifically, the proposed framework captures the spatial dependency using bidirectional random walks on the graph, and the temporal dependency using the encoder-decoder architecture with scheduled sampling and the sequential modeling from the nature of recurrent neural network.**

**Rating:** 9
**Confidence:** 5

**Review:**

Summary of the reviews:
Pros:
•	A Novel Diffusion Convolutional Recurrent Neural Network framework for traffic forecasting
•	Apply bidirectional random walks with nice theoretical analysis to capture the spatial dependency
•	Novel applications of sequence to sequence architecture and the scheduled sampling technique into modeling the temporal dependency in the traffic domain
•	This work is very well written and easy to follow
Cons:
•	Needs some minor re-organization of contents between the main sections and appendix

Detailed comments:
D1:  Some minor re-organization of contents between the main sections and the appendix will help the reader reduce cross-section references. Some examples:
1.	Lemma 2.1 can be put into appendix since it is not proposed by this work while the new theoretical analysis of Appendix 5.2 (or at least a summary) can be moved to the main sections
2.	Figure 9 can be moved earlier to the main section since it well supports one of the contributions of the proposed method (using the DCRNN to capture the spatial correlation)
D2: Some discussions regarding the comparison of this work to some state-of-the-arts graph embedding techniques using different deep neural network architectures would be a plus

---

> ### Author Response · Authors · 2017-12-28
> **Response to ICLR 2018 Conference Paper804 AnonReviewer2**
>
> We sincerely thank the reviewer for the detailed and constructive comments. We will follow the suggestions to reorganize the content and add more discussions about the relationship between the proposed approach and state-of-the-art deep neural network based graph embedding techniques.

---

### Author Response · Authors · 2018-02-25
**Revision Summary**

We have uploaded the camera-ready version of the paper with mainly the following changes:
- Adds discussion of DeepTransport [1], Gaussian Processes [2] and Spatial KNN [3]  in Section 4 and Appendix D.
- Adds discussion of graph embedding in Appendix D.
- Adds explanation for the efficient calculation of Equation 2 in Appendix B.
- Adds more description of the datasets in Appendix E.1.
- Adds URL for source code: https://github.com/liyaguang/DCRNN
- Adds acknowledgment.
- Fixes typos.

[1] X. Cheng et al. DeepTransport: Learning Spatial-Temporal Dependency for Traffic Condition Forecasting.
[2] Y. Xie et al. Gaussian Processes for Short-Term Traffic Volume Forecasting.
[3] P. Cai et al. A spatiotemporal correlative k-nearest neighbor model for short-term traffic multi-step forecasting.

---

### Decision · Program_Chairs · 2018-01-29
**ICLR 2018 Conference Acceptance Decision**

**Decision:**

Accept (Poster)

**Comment:**

The paper received highly diverging scores: 5 (R1) ,9 (R2), 4(R3). Both R1 and R3 complained about the comparisons to related methods. R3 suggested some kNN and GP baselines, while R1 mentioned concurrent work using deepnets for trafffic prediction.

R3 is real expert on field. R2 and R1, not so.
R2 review very positive, but vacuous.
Rebuttal seems to counter R1 and R3 well.

It's a close all but the AC is inclined to accept since it's an interesting application of (graph-based) deepnets.